# An Improved Differential Evolution Adaptive Fuzzy PID Control Method for Gravity Measurement Stable Platform

**DOI:** 10.3390/s23063172

**Published:** 2023-03-16

**Authors:** Xin Chen, Hongwei Bian, Hongyang He, Fangneng Li

**Affiliations:** School of Electrical Engineering, Naval University of Engineering, Wuhan 430033, China

**Keywords:** gravity measurement, stable platform, adaptive control, PID control

## Abstract

In the platform gravimeter, the stabilization accuracy of the gravimetric stabilization platform is crucial to improve the accuracy of gravity value measurements due to its uncertainties, such as mechanical friction, inter-device coupling interference, and nonlinear disturbances. These cause fluctuations in the gravimetric stabilization platform system parameters and present nonlinear characteristics. To resolve the impact of the above problems on the control performance of the stabilization platform, an improved differential evolutionary adaptive fuzzy PID control (IDEAFC) algorithm is proposed. The proposed enhanced differential evolution algorithm is used to optimize the initial control parameters of the system adaptive fuzzy PID control algorithm to achieve accurate online adjustments of the gravimetric stabilization platform’s control parameters when it is subject to external disturbances or state changes and attain a high level of stabilization accuracy. The results of simulation tests, static stability experiments, and swaying experiments on the platform under laboratory conditions, as well as on-board experiments and shipboard experiments, all show that the improved differential evolution adaptive fuzzy PID control algorithm has a higher stability accuracy compared with the conventional control PID algorithm and traditional fuzzy control algorithm, proving the superiority, availability, and effectiveness of the algorithm.

## 1. Introduction

The Earth’s gravitational field is one of the essential fundamental physical fields of the Earth [1,2]. Gravitational field data are national strategic data that are irreplaceable in basic mapping, national defense and military, resource exploration, earth science, aerospace, and national integrated PNT system construction [3]. Accurate gravity measurements are obtained with the help of gravimeters, which can be divided into platform type [4,5] and strapdown type [6,7]. The platform-type gravimeters primarily consist of a gravity sensor and stabilization platform, where the gravity sensor is responsible for the accurate measurement of gravity value information and ensuring that it has a stable vertical pointing during its movement. This requires the use of a gravity measurement stabilization platform, and the installation relationship between the two is shown in Figure 1.

As shown in Figure 1, the horizontal attitude error of stable gravity measurements is θ, and the vertical deviation of the sensitive axis of the gravity sensor is also θ. In the figure, g represents the actual value of gravity, g′ represents the gravity value sensed by the gravity sensor, $a_{h}$ represents the horizontal acceleration of the carrier, and $a_{h}\prime$ represents the influence of the horizontal acceleration of the page on the measured value of the gravity sensor.

Then, the gravity measurement error can be expressed as:(1)$$\begin{array}{l} e_{g}=g\prime -a_{h}\prime -g\\ \;\;\;\;=g\cos\theta -a_{h}\sin\theta -g\\ \;\;\;\;\approx -a_{h}\cdot \theta -g\cdot \theta^{2}/2 \end{array}$$

In Equation (1), the second term is a steady-state error, which exists even if the carrier does not move.

If the gravity static measurement accuracy of 1 mGal is to be achieved, the horizontal misalignment angle must be less than 4.8′ when the first item in Equation (1) is ignored. The magnitude of the first term in Equation (1) depends on the horizontal acceleration of the carrier and the extent of the error angle of the platform. If the gravity sensor’s damping mechanism is not considered, the maximum dynamic gravity measurement error should not exceed 1 mGal. When the horizontal acceleration of the carrier is $a_{h}=0.1\;m/s^{2}$, the error angle θ should be less than 20.6″. In actual measurement systems, the damping mechanism, such as silicone oil, is set in the gravity sensor, which gives the gravity measurement system a specific anti-interference ability against the horizontal acceleration of the carrier. In practical applications, the error angle θ does not need to be less than 20.6″. However, Equation (1) indicates that, in the case of the horizontal misalignment angle of the gravimetric measuring platform, the horizontal acceleration of the carrier has a non-negligible effect on the gravimetric measurement results.

As large-weight instrumentation, a gravimeter requires a gravimetric stabilization platform with a hefty load capacity (more than 30 kg). The related research mainly studies the control algorithm of a small load, three-axis or airborne stabilization platform. There is less research on the control of high-precision stabilization platforms with enormous load capacities. Domestic and foreign research scholars have conducted relevant studies to reduce the stabilized platform’s control system error and improve the stabilized platform’s stabilization accuracy. In Ref. [8], a robust adaptive controller in the form of expectation compensation was proposed for accurate tracking control, to reduce the influence of measurement noise and to improve the control performance of the photoelectric gyro-stabilized platform system. In Ref. [9], a new decoupling algorithm, applied to the stabilized platform for aerial remote sensing, was used to solve the problems of unknown nonlinear interference and coupling interference, which occur in tasks such as aerial photography. In Ref. [10], a combination of sliding mode control and a neural network is proposed to deal with the uncertainty disturbances in the inertially stabilized platform (ISP) system model. The article also presents an adaptive neural network to approximate the uncertainty and unknown disorders of the system and improve control performance. An anti-disturbance control scheme for an airborne radar stabilization platform based on the inverse estimation algorithm of self-rejecting control (ADRC) is used to solve the stability and clarity problems of radar imaging [11]. An improved dynamic variational differential evolution algorithm (DMDE) is proposed to optimize the PID control parameters and is validated by simulation for several standard industrial-controlled object models [12]. An adaptive fuzzy PID composite control algorithm is proposed and applied to control gyro-stabilized platforms for resisting nonlinear disturbance factors in platform control [13]. A fuzzy PID composite control is proposed to improve stability and resistance to disturbances in aero-inertial stabilized platforms [14]. A fuzzy PID intelligent control algorithm is proposed to solve the low control efficiency of small stable platforms due to frame coupling [15]. Most of the control algorithms proposed in these studies have only been tested in simulations. They are rarely applied to existing stabilized platforms, so studying an intelligent control algorithm that can be used in actual gravimetric stabilized platforms is crucial.

## 2. System Description

### 2.1. System Architecture

The gravimetric stabilization platform in this paper was developed in the Marine/Aerospace Gravimeter project, which is a primary national science and technology project with a two-axis structure (pitch axis and roll axis). The mechanical structure is shown in Figure 2.

The relevant coordinate system is defined as follows [16].
Inertial coordinate system i: The inertial coordinate system has the center of the Earth as the origin, and the $x_{i}$ and $y_{i}$ axes are in the equatorial plane of the Earth, where the $x_{i}$ and $z_{i}$ axes are set to point from the head to the equinox and along the Earth’s rotation axis to the North Pole direction, respectively.Earth coordinate system e: The Earth coordinate system has the center of the Earth as the origin, and the $x_{e}$ and $y_{e}$ axes are in the Earth’s equatorial plane, where the $x_{e}$ and $z_{e}$ axes are set to point from the head to the prime meridian and along the Earth’s rotation axis to the North Pole direction, respectively.Geographic coordinate system n: The geographic coordinate system takes the center of mass of the carrier as the origin; $x_{n}$, $y_{n}$ and $z_{n}$ axes are set to point east, north, and skyward from the head, respectively.Carrier coordinate system b: The carrier coordinate system takes the center of mass of the carrier as the origin, and the $x_{b}$, $y_{b}$, and $z_{b}$ axes are set to point to the right side of the carrier, to the front of the carrier, and the top of the page, respectively, and form a right-handed coordinate system.Outer frame (roll frame) coordinate system r: set the direction of $y_{r}$ axes to be the same as the direction of the $y_{b}$ axes, and its coordinate system can only rotate around the $y_{b}$ axes during the subsequent movements relative to the coordinate system b. The resulting roll angle is set to $\gamma_{r}$.The inner frame (pitch frame) coordinate system f: set the direction of the $x_{f}$ axes to be the same as the pointing of the $x_{b}$ axes, and its coordinate system can only rotate around the $x_{r}$ axes during the subsequent motions relative to the coordinate system r. The resulting pitch angle is set to $\theta_{f}$.

### 2.2. Principle of Isolation Carrier Movement

The structure of the gravimetric stabilization platform was designed so that the IMU is fixed directly under the gravimeter base. When the carrier moves or is otherwise disturbed, the inner and outer frames will shake slightly, resulting in the stabilization platform not coinciding with the local geographic level. Let $\omega_{nb}^{b}={\left[\omega_{nbx}^{b}\;\omega_{nby}^{b}\;\omega_{nbz}^{b}\right]}^{T}$ be the projection of the angular velocity of the carrier coordinate system relative to the geographic coordinate system on the carrier coordinate system; the pitch axis compensation angular velocity generated by the torque motor in the stabilization loop is ${\dot{\theta }}_{f}$, and the roll axis compensation angular velocity is ${\dot{\gamma }}_{r}$. Since the angular velocity output from the IMU is the angular velocity of the gravimeter relative to the inertial space, the angular velocity $\omega_{ib}^{b}$ cannot be obtained by direct measurement from $\omega_{nb}^{b}=\omega_{ib}^{b}-\omega_{in}^{b}=\omega_{ib}^{b}-C_{n}^{b}\omega_{in}^{n}$, where $C_{n}^{b}$ denotes the attitude array of the coordinate navigation system b relative to the carrier system n, $\omega_{ib}^{b}$ is the gyro output, and $\omega_{in}^{n}$ indicates the rotation of the n system close to the i system. This contains two components: the wheel of the navigation system caused by the course of the Earth, and the process of the inertial guidance system moving near the surface of the Earth due to the n-system process caused by the bending of the Earth’s surface. Accordingly, $\omega_{in}^{n}=\omega_{ie}^{n}+\omega_{en}^{n}$.

where $\omega_{ie}^{n}={\left[0\;\omega_{ie}\cos L\;\omega_{ie}\sin L\right]}^{T}$,
$$\omega_{en}^{n}={\left[\begin{array}{ccc} -\frac{v_{N}}{R_{M}+h} & \frac{v_{E}}{R_{N}+h} & \frac{v_{E}}{R_{N}+h}\tan L \end{array}\right]}^{T}$$
where $\omega_{ie}$ is the angular rate of Earth’s rotation; L and h are the geographic dimension and altitude, respectively; $v_{N}$ and $v_{E}$ are the northward and eastward velocities, respectively; $R_{M}$ and $R_{N}$ are the radius of principal curvature of the meridian circle and the radius of principal curvature of the Uranus circle, respectively.

The angular velocity of the transverse roll axis for the geographical coordinate system is obtained based on the relevant coordinate transformations.
(2)$$\begin{array}{ll} \omega_{nr}^{r} & =C_{b}^{r}\omega_{nb}^{b}+\omega_{br}^{r}\\ & =\left[\begin{array}{ccc} \cos\gamma_{r} & 0 & \sin\gamma_{r}\\ 0 & 1 & 0\\ -\sin\gamma_{r} & 0 & \cos\gamma_{r} \end{array}\right]\left[\begin{array}{c} \omega_{nbx}^{b}\\ \omega_{nby}^{b}\\ \omega_{nbz}^{b} \end{array}\right]+\left[\begin{array}{c} 0\\ {\dot{\gamma }}_{r}\\ 0 \end{array}\right]\\ & =\left[\begin{array}{c} \omega_{nbx}^{b}\cos\gamma_{r}+\omega_{nbz}^{b}\sin\gamma_{r}\\ \omega_{nby}^{b}+{\dot{\gamma }}_{r}\\ -\omega_{nbx}^{b}\sin\gamma_{r}+\omega_{nbz}^{b}\cos\gamma_{r} \end{array}\right] \end{array}$$
where $\omega_{nr}^{r}$ is the projection of the angular velocity of the rotation of the r system to the n system on the r system; $C_{b}^{r}$ is the coordinate system transformation matrix from the r system to the b system; $\omega_{nb}^{b}$ is the projection of the angular velocity of the rotation of the b system on the n system on the b system; $\omega_{br}^{r}$ is the projection of the angular velocity of the rotation of the r system to the rotation of the b system on the r system; $\gamma_{r}$ is the roll angle; ${\dot{\gamma }}_{r}$ is the rolling angular velocity; $\omega_{nbx}^{b}$, $\omega_{nby}^{b}$, and $\omega_{nbz}^{b}$ are the components of the projection of the angular velocity of the rotation of the b system to the n system on the b system on the x, y and z axes, respectively.

Similarly, the angular velocity of the pitch axis for the geographical coordinate system can be obtained.
(3)$$\begin{array}{ll} \omega_{nf}^{f} & =C_{r}^{f}\omega_{nr}^{r}+\omega_{rf}^{f}\\ & =\left[\begin{array}{ccc} 1 & 0 & 0\\ 0 & \cos\theta_{f} & -\sin\theta_{f}\\ 0 & \sin\theta_{f} & \cos\theta_{f} \end{array}\right]\left[\begin{array}{c} \omega_{nrx}^{r}\\ \omega_{nry}^{r}\\ \omega_{nrz}^{r} \end{array}\right]+\left[\begin{array}{c} {\dot{\theta }}_{f}\\ 0\\ 0 \end{array}\right]\\ & =\left[\begin{array}{c} \omega_{nbx}^{b}\cos\gamma_{r}+\omega_{nbz}^{b}\sin\gamma_{r}+{\dot{\theta }}_{f}\\ \left(\omega_{nby}^{b}+{\dot{\gamma }}_{r}\right)\cos\theta_{f}+A\cdot \sin\theta_{f}\\ \left(\omega_{nby}^{b}+{\dot{\gamma }}_{r}\right)\sin\theta_{f}-A\cdot \cos\theta_{f} \end{array}\right] \end{array}$$
where $\begin{array}{r} A=\omega_{nbx}^{b}\sin\gamma_{r}-\omega_{nbz}^{b}\cos\gamma_{r} \end{array}$$\omega_{nf}^{f}$ is the projection of the angular velocity of the rotation of the f system to the n system on the f system; $C_{r}^{f}$ is the coordinate system transformation matrix from the f system to the r system; $\omega_{nr}^{r}$ is the projection of the angular velocity of the rotation of the r system on the n system on the r system; $\omega_{rf}^{f}$ is the projection of the angular velocity of the rotation of the f system to the rotation of the r system on the f system; $\theta_{f}$ is the pitch angle; ${\dot{\theta }}_{f}$ is pitch angular velocity; $\omega_{nrx}^{r}$, $\omega_{nry}^{r}$, and $\omega_{nrz}^{r}$ are the components of the projection of the angular velocity of the rotation of the r system to the n system on the r system on the x, y and z axes, respectively.

To maintain the gravimetric stabilization platform plane at all times within the local geographic level, the angular velocity of the inner frame relative to the geographic coordinate system must be 0 after the initial righting, and the following conditions must be met.
(4)$$\{\begin{array}{c} \omega_{nfx}^{f}=0\\ \omega_{nfy}^{f}=0 \end{array}$$

Equations (3) and (4) can be obtained by the two torque motors to produce the angular speed, meeting the following conditions.
(5)$$\{\begin{array}{l} {\dot{\theta }}_{f}=-\omega_{nbz}^{b}\sin\gamma_{r}-\omega_{nbx}^{b}\cos\gamma_{r}\\ {\dot{\gamma }}_{r}=\left(-\omega_{nbx}^{b}\sin\gamma_{r}+\omega_{nbz}^{b}\cos\gamma_{r}\right)\tan\theta_{f}-\omega_{nby}^{b} \end{array}$$

When the control result satisfies Equation (5), the gravimetric stabilization platform table surface will remain parallel to the local geographical level.

### 2.3. Control System Block Diagram

The servo control system is the core of the gravimetric stabilization platform. Additionally, it ensures algorithm implementation and platform stability accuracy, so a scientific and reasonable control system structure is essential. The gravity measurement stable platform’s hardware structure consists of a pitch axis and a roll axis with the same design, without considering the influence of coupling factors between the two axes. Only one of the axes can be analyzed when analyzing its control law, and the pitch axis is used as an example in the text with no further explanation. Figure 3 depicts the system composition and the principle block diagram.

Given that the stabilized platform must install heavy equipment such as an IMU and a gravimeter and has stringent real-time requirements, its control system employs a multi-stage composite control scheme, which utilizes dual position and velocity rings to achieve composite control. Figure 4 depicts the hybrid control structure diagram.

Where $\theta_{d}$ is the pitch axis target angle, $\dot{\theta }$ is the angular velocity, θ is the actual output pitch axis angle, the load includes the gravimeter and the IMU fixedly connected.

## 3. Improved Differential Evolutionary Adaptive Fuzzy PID Control Algorithm

The gravimetric stabilized stage is a servo-controlled system with strong nonlinearity. Due to its uncertainties, such as random disturbances, friction, mechanical coupling, and gyroscopic drift, the exact mathematical model of the stabilized platform cannot be determined. In contrast, fuzzy control is a kind of control based on fuzzy rules, which is used to realize the control of a nonlinear system without a precise mathematical model [17]. Figure 5 depicts the structure of a typical fuzzy control system.

Fuzzy control can adjust the torque motor’s control voltage according to the fuzzy rules. However, its steady-state accuracy generally does not meet the accuracy requirements of the gravimetric stabilization platform. Therefore, the control of the stabilized platform is usually realized by the PID controller, which can achieve high control and stabilization accuracy with good real-time accuracy and strong robustness when it has suitable control parameters. To meet the requirements of high precision and robust system stability, a combination of fuzzy control and PID control can realize the real-time adjustment of the control parameters using the control system according to the error and the differential of the error. Then, the adaptive fuzzy PID control of the system can be determined.

The accuracy of the initial parameter-setting of adaptive fuzzy PID control significantly impacts the control performance. In the conventional adaptive fuzzy PID control, the selection of initial control parameters is often determined based on experience or traditional parameter tuning methods, and its accuracy is inadequate. The differential evolution algorithm is an intelligent algorithm that can perform automatic optimization searches. An adaptive differential evolution algorithm based on a multivariate strategy with variable parameters (ADE-MMVP) is attempted to optimize the initial control parameters of an adaptive fuzzy PID control and realize the accurate control of a stable platform.

### 3.1. ADE-MMVP Algorithm

Differential Evolution (DE) was proposed by Professor Rainer Storn and Professor Kenneth Price in 1995 [18,19]. DE is mainly implemented through competition and selection among individuals in the population to optimize the population. It is a global search algorithm. The differential evolution algorithm adopts an accurate number coding and is based on the differential strategy mutation operation, as in Ref. [20], which can effectively reduce genetic complexity. At the same time, the algorithm also has a specific memory ability, so the global search ability of the algorithm is greatly improved, and the robustness is enhanced. In general, the differential evolution algorithm has the advantages of fewer undetermined parameters, a fast convergence speed, and the ability to easily obtain the global optimum. Compared with other optimization algorithms, the differential evolution algorithm has the following characteristics: [21].
(1)The ability to deal with nonlinear, non-differentiable, and multimodal functions;(2)The ability to process intensive cost functions in parallel;(3)It is easy to use and can realize algorithm optimization while making good use of control variables;(4)Good convergence, which can converge to the optimal global value in continuous independent tests.

Due to the single variation strategy, fixed variation factor, and crossover probability factor of the conventional differential evolution algorithm, the algorithm is prone to slow convergence and failure to converge to the global optimum. To solve this problem, we propose an ADE-MMVP algorithm with multiple variation strategies, various factors, and crossover probability factors that can adaptively change with population evolution [22].
Variation strategy of ADE-MMVP algorithm

Among all DE variation strategies, DE/rand/1 is the most widely used and most beneficial for maintaining population diversity, as in Refs. [23,24] DE/best/2 has the optimal solution, which beneficial to resolving some technical problems with the algorithm and accelerate the convergence speed, but the variation strategy containing optimal information is more likely to fall into local optimum [25]. Based on this, the two variational strategies are combined using a certain ratio and adjusted by a scaling factor, and then adapted in a variety of application scenarios. The specific calculation process is as follows.
(6)$$h_{i}^{1}(t+1)=x_{r1}(t)+F(x_{r2}(t)-x_{r3}(t))$$
(7)$$h_{i}^{2}(t+1)=x_{\mathrm{best}}(t)+F(x_{r1}(t)-x_{r2}(t))+F(x_{r4}(t)-x_{r5}(t))$$
(8)$$h_{i}(t+1)=\lambda \times h_{i}^{1}(t+1)+(1-\lambda )\times h_{i}^{2}(t+1)$$
where t is the current number of iterations, $x_{r1}(t)$~$x_{r5}(t)$ are the randomly selected mutually dissimilar individuals in the initial population, $x_{i}(t)$ is the current individual, $x_{\mathrm{best}}(t)$ is the current optimal individual, F is the variation factor, λ is the proportion factor of DE/rand/1 in the variation strategy.
2.Variation factor of ADE-MMVP algorithm

The variation factor F mainly controls the search step size of the differential evolution algorithm, which affects the algorithm population’s diversity and convergence. The algorithm uses a variation factor that dynamically adjusts with the number of iterations.
(9)$$x(G)=e^{1-\frac{G_{m}}{G_{m}+1-G}}$$
(10)$$F(G)=F_{\min}+x(G)(F_{\max}-F_{\min})$$
where G denotes the current number of iterations and the maximum number of iterations, $G_{m}$ denotes the value of the variable factor of the current number of iterations, and $F_{\max}$ and $F_{\min}$ represent the maximum and minimum values of the variation factor, respectively.
3.Crossover probability factor of ADE-MMVP algorithm

The crossover probability factor CR controls the degree of participation of randomly selected individuals in the algorithm crossover, thus balancing the role of the local search and global search relationship. The following equation adjusts the algorithm crossover probability factor CR.
(11)$$CR=CR_{\min}+\frac{\left(CR_{\max}-CR_{\min}\right)}{G_{\max}}G$$
where $G_{\max}$ denotes the maximum number of iterations, $CR_{\min}$ denotes the minimum value of the set crossover probability factor, $CR_{\max}$ denotes the maximum value of the set crossover probability factor, and G denotes the current number of iterations.

### 3.2. Adaptive Fuzzy PID Control Algorithm with ADE-MMVP Optimization

To solve the problem regarding the accuracy of the initial control parameters of the adaptive fuzzy PID controller, [26] ADE-MMVP is used for automatic optimization of the control parameters, and the optimization results are transmitted to the adaptive fuzzy PID controller. The system uses a dual-loop velocity loop and position loop to achieve composite control.

In the platform mathematical model, the highest numerator and denominator of the speed loop transfer function are the same, and PI control is used in the speed loop control system to ensure the system’s stable operation. The structure of the speed loop control system is shown in Figure 6.

Structure of the position loop control system is shown in Figure 7.

Set several rules for PID controller control parameter adjustment according to the control system requirements.
When the system error |e| is large, a more significant proportional link coefficient should be used to speed up the response of the system. However, the proportional coefficient cannot be infinite or the system will have a massive amount of overshoot and the adjustment time of the system will increase, as the system error |e| is significant at the beginning of the system. To avoid the system control exceeding the maximum execution range of the actuator, a smaller differential coefficient should be taken at this time to speed up the system response. To avoid causing a massive overshoot to the system, the integral link should be removed when the error |e| is large and the integral coefficient $K_{i}=0$ is taken.When the system error |e| is moderately large, the proportional link coefficient should be appropriately reduced to take a minor $K_{p}$. This will prevent the system from having a massive amount of overshoot, resulting in system collapse. The response speed of the system and the value of the differential link are directly related, so the value of differential link coefficient $K_{d}$ at this time is critical. The magnitude of the integral link coefficient can be increased in this scenario, but this increase should not be too tremendous.When the system error |e| is minor, to ensure the system has a good steady-state performance, this time can use a more significant differential link coefficient and integral link coefficient. To avoid system oscillation, the value of $K_{d}$ should be obtained appropriately.

### 3.3. Simulation and Analysis

Table 1 displays the ADE-MMVP algorithm parameter settings used in the simulation trials. The rectification outcomes of the ZN critical proportionality technique define the initial parameter values for the adaptive fuzzy control algorithm (AFC). In contrast, the improved differential evolution adaptive fuzzy PID control’s (IDEAFC) initial control parameters are established by following the best control parameters, as identified by the ADE-MMVP method.

The response of the speed loop under the two signals is given in Figure 8, which depicts the simulation of the speed and position loops using the sinusoidal and sawtooth wave signals, respectively. Figure 9 displays the position loop’s reaction to the two movements.

The experimental results show that the IDEAFC algorithm is superior.

## 4. Platform Experiments

Based on the simulation test results, this chapter presents the platform experiments that were conducted to further verify the algorithm’s superiority. The stable platform control system had a TMS320F28335 digital processor as the control core. The traditional PID control algorithm, AFC algorithm, and IDEAFC algorithm were used in laboratory static and dynamic rocking experiments (longitudinal and transverse rocking experiments, respectively). The electronic self-collimator and IMU data were collected to control each algorithm. The effect of each algorithm was compared by collecting data from the electronic self-collimator and IMU. The control effects were compared by collecting data from the electronic self-collimator and the IMU. The IDEAFC algorithm was also used for onboard experiments and shipboard experiments, and the control effects were analyzed using the data collected from the IMU.

### 4.1. Shaking Table Static Stability Experiment

The gravimetric stabilization platform was subjected to a shaking table static stabilization experiment in a laboratory environment. The angular offsets in the investigation were collected from IMU attitude data and TriAngle electronic self-collimator data, respectively. The physical objects of IMU and the electronic self-collimator are shown in Figure 10; the experimental environment of the platform laboratory is shown in Figure 11.

In the static experiment, the angle measured by the electronic self-collimator is the angle deviation value of the overall stabilized platform, which can truly reflect the overall accuracy of the forum. In contrast, the angle deviation value measured by the IMU can reflect the control accuracy of the control algorithm. GNSS positioning-assisted calibration was used throughout the experiment to avoid the influence of measurement errors caused by the IMU’s own drift in the test results.

When running the tests, multiple experiments were conducted, utilizing the conventional PID method, the AFC algorithm, and the IDEAFC algorithm. The set closest to the mean value was chosen as the outcome. Figure 12 depicts the IMU pitch and roll angle data gathered throughout the studies.

As seen from the comparison in Figure 12, in the experimental results obtained by IMU under static conditions, whether pitch angle or roll angle, the control effect of the IDEAFC algorithm was shown to be better than the other two. The standard deviation of the pitch angle of the IDEAFC algorithm was 0.033″, with a peak value of 0.002°; the standard deviation of the roll angle was 1.5″, with a peak value of 0.002°.

The data collected by the electronic self-collimator and IMU were compared separately to study the relationship between the overall stability accuracy and control accuracy of the platform. Since the electronic self-collimator is a high-precision optical goniometer, its reflector is small and fixed, so it can only be used under static test conditions. The relationship between the platform’s overall stability and control accuracy were evaluated from the comparison experiment under static conditions, and then used to analyze whether the stability accuracy of various algorithms meets the requirements.

In the static experiment, the pitch angle and the roll angle measured by the electronic self-collimator are shown in Figure 13.

A comparison of the control effects of the three control algorithms can be obtained from the comparison chart, as shown in Table 2.

The improvement in accuracy was calculated as follows.

Relative comparison algorithm precision improvement ratio = (comparison algorithm precision − target algorithm precision)/comparison algorithm precision.

### 4.2. Dynamic Rocking Experiment

The dynamic swaying experiment of the gravity measurement stable platform was conducted in the laboratory using a swaying table with adjustable longitudinal and transverse swaying amplitudes. The maximum value of longitudinal swaying was ±45°, and the total value of transverse swaying was ±30°. In the natural gravity measurement environment, whether vehicle-mounted, on the lake, at sea, or under the sea, the swaying angle generally did not exceed ±10°. To test the stability of the platform, when conducting the platform swaying experiment, the angle of both longitudinal and transverse swaying was set to ±15°. At the same time, to avoid the impact caused by the different swing speeds, the multi-grade swing experiment was conducted separately. The swing speed of each gear was different, and the period setting for each pack is shown in Table 3.

Figure 14 compares pitch angle data and roll angle data at different gears during vertical rocking.

The comparison chart shows that the control effect of the algorithm is the same under different shaking table operation gears, so the impact of other gears on the control algorithm were not considered in the subsequent comparison test.

A comparison of the data measured by IMU under the three control algorithms during the longitudinal rocking test is shown in Figure 15.

The comparison graphs show the control effects of the three control algorithms in the longitudinal rocking experiment, as shown in Table 4.

The comparison of data under the three control algorithms was measured by IMU during the swing test, as shown in Figure 16.

Using the comparison graphs, we can compare the control effects of the three control algorithms in the swing experiment, as shown in Table 5.

### 4.3. In-Vehicle Experiments

The gravimetric stabilization platform was tested in-vehicle to verify the stability performance of the IDEAFC algorithm in real application scenarios. The designed route in the experiment was Wuhan city road, the vehicle driving speed was 30 km/h, and an onboard experimental environment was built, as shown in Figure 17.

The vehicle-based experimental driving city route was as shown in Figure 18.

The pitch and roll angle data were collected during the experiment, as shown in Figure 19.

During the experiment, the standard deviation of the pitch angle of the gravimetric stabilized platform was 7.56″, and the standard deviation of the roll angle was 7.09″, both within 10″.

### 4.4. Shipboard Experiments

A gravimetric stabilization platform using an IDEAFC algorithm was used for a boat experiment conducted at Mulan Lake, Wuhan, using a small electric pleasure boat. The boat is shown in Figure 20 while docked at the experimental dock. GNSS was used to calibrate the inertial guidance during the experiment, and the GNSS antenna was installed directly above the IMU on the boat top platform, as shown in Figure 21.

The experimental waters contained light waves, with a wind speed of about 1 m/s; the speed of the boat in the lake was 15 km/h. From the docked experimental dock, sailing experiments were conducted around Mulan Lake; the whole route map is shown in Figure 22.

The pitch and roll angle data obtained throughout the cruise are shown in Figure 23.

During the whole shipboard experiment, the standard deviation of the gravity measurement stabilized platform pitch angle was 2.11″, and the standard deviation of the roll angle was 3.69″, both within 10″.

## 5. Conclusions

An improved differential evolutionary adaptive fuzzy PID control method was proposed to reduce the influence of multi-source perturbations on the gravimetric stabilization platform, achieve high precision control, and improve stabilization accuracy. As a comparison, conventional PID and adaptive fuzzy control were also applied to the platform control. Simulation experiments and platform tests were carried out for different algorithms. The experimental results show that the improved differential evolution algorithm can significantly improve the stabilization accuracy of the gravimetric stabilization platform. In the laboratory swing experiments, the control accuracy was enhanced by more than 25% under static conditions and more than 18% under dynamic conditions. In the onboard and shipboard experiments, the improved differential evolution adaptive fuzzy PID control algorithm pitch and roll angle standard deviation were within 10″. This lays a foundation for subsequent gravity measurements under static or dynamic conditions and creates the necessary conditions for the realization of high-precision dynamic gravity measurements.

## Figures and Tables

**Figure 1 sensors-23-03172-f001:**
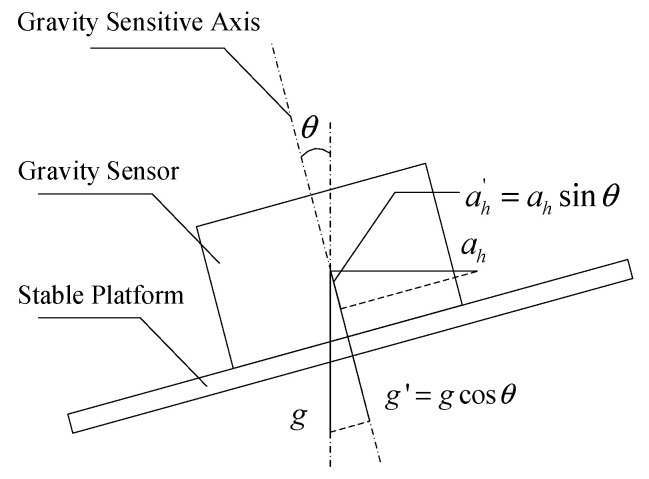
Gravity measurement of stable platform and gravity sensor installation relationship.

**Figure 2 sensors-23-03172-f002:**
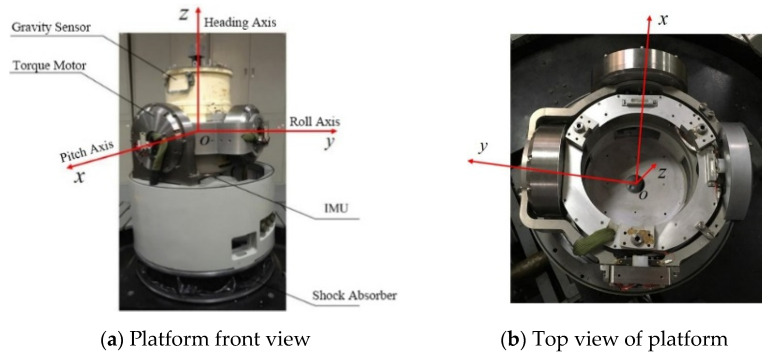
Stabilization platform mechanical structure.

**Figure 3 sensors-23-03172-f003:**
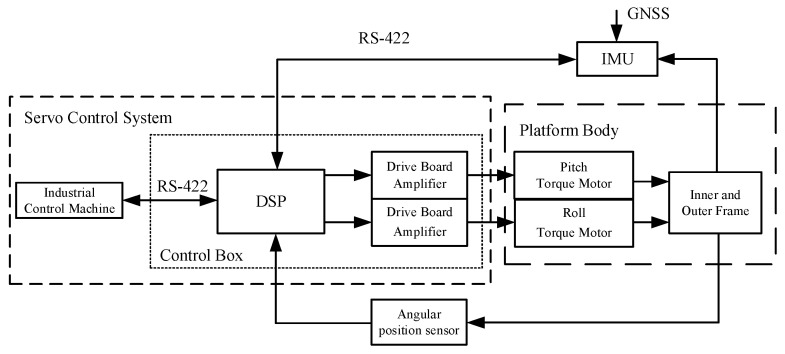
Stabilized platform system composition and the principle block diagram.

**Figure 4 sensors-23-03172-f004:**
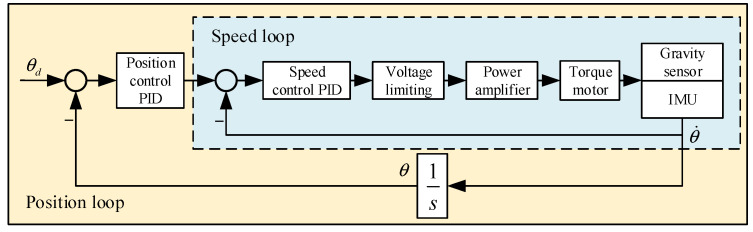
The hybrid control structure diagram.

**Figure 5 sensors-23-03172-f005:**
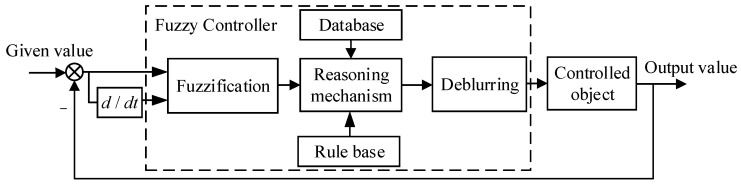
Typical fuzzy control system structure.

**Figure 6 sensors-23-03172-f006:**
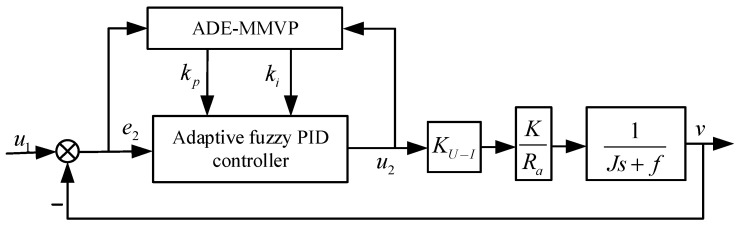
Structure of speed loop control system.

**Figure 7 sensors-23-03172-f007:**
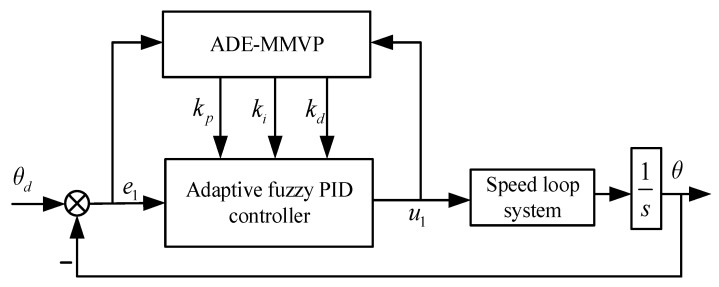
Structure of the position loop control system.

**Figure 8 sensors-23-03172-f008:**
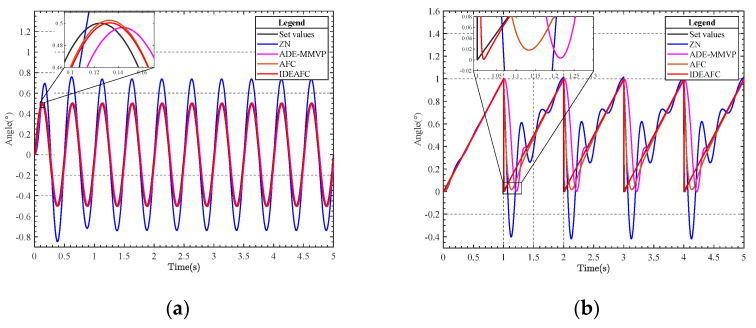
(**a**) Comparison curve of sine signal response of speed ring; (**b**) comparison curve of sawtooth wave response of speed ring gravimeter.

**Figure 9 sensors-23-03172-f009:**
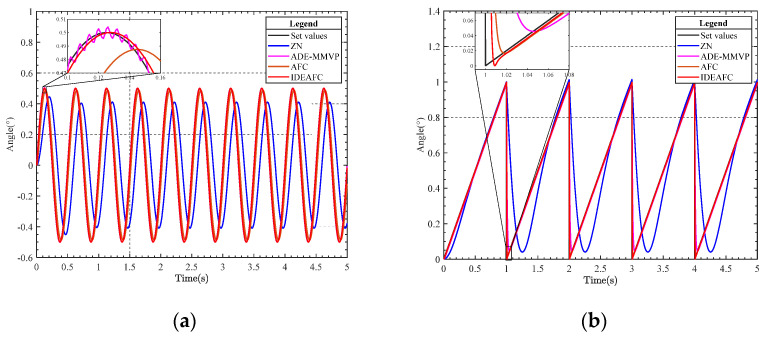
(**a**) Comparison curve of sine signal response of location ring; (**b**) comparison curve of sawtooth wave response of location ring.

**Figure 10 sensors-23-03172-f010:**
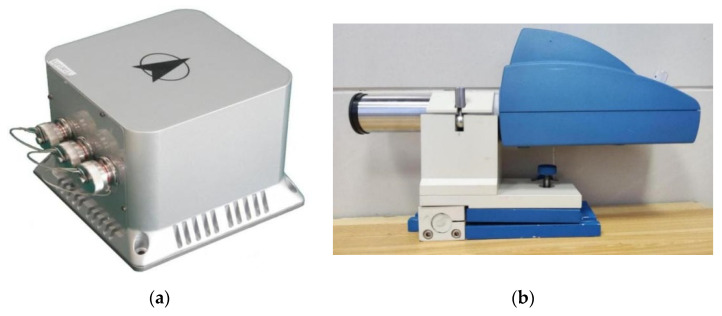
(**a**) IMU; (**b**) electronic autocollimator.

**Figure 11 sensors-23-03172-f011:**
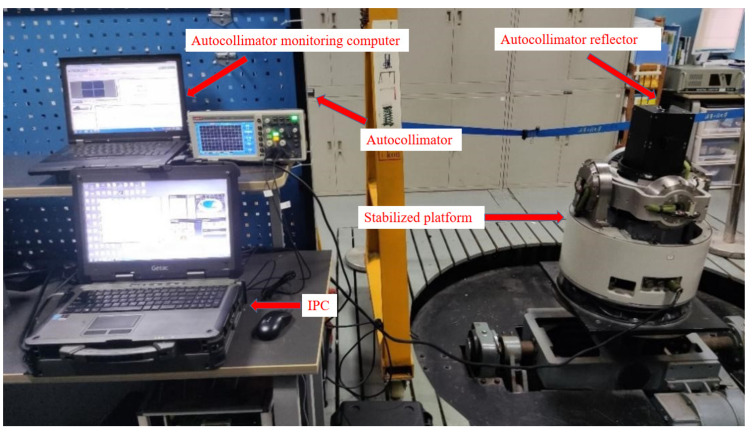
The experimental environment of the platform laboratory.

**Figure 12 sensors-23-03172-f012:**
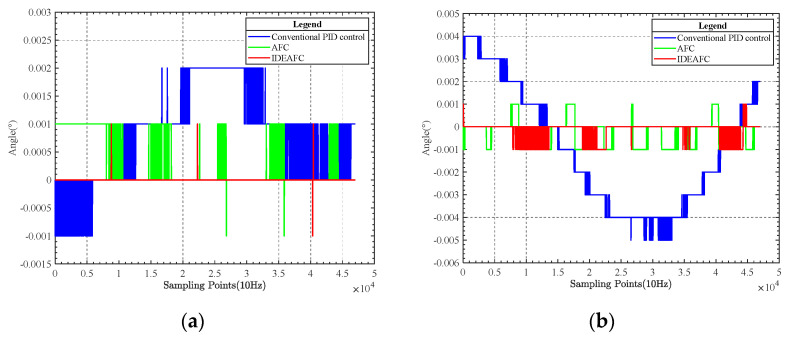
(**a**) Comparison of pitching angle data; (**b**) comparison of rolling angle data.

**Figure 13 sensors-23-03172-f013:**
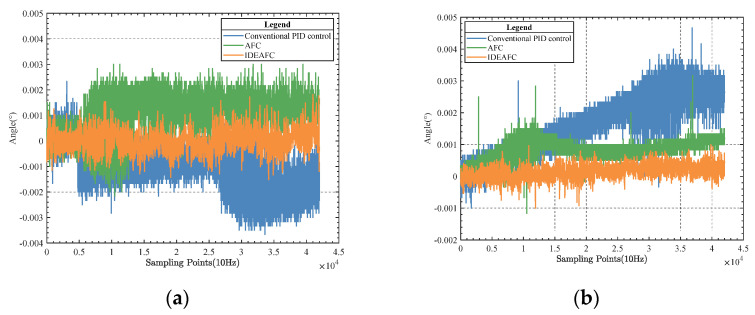
(**a**) Comparison of static pitch angle data; (**b**) comparison of static roll angle data.

**Figure 14 sensors-23-03172-f014:**
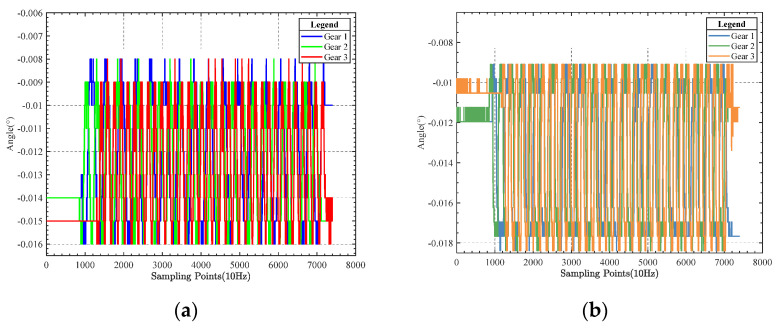
(**a**) Dynamic experimental pitch angle data; (**b**) dynamic experimental roll angle data.

**Figure 15 sensors-23-03172-f015:**
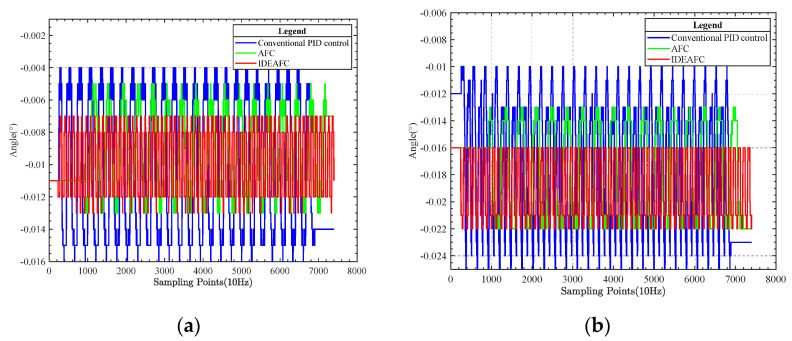
(**a**) Pitch angle data comparison; (**b**) roll angle data comparison.

**Figure 16 sensors-23-03172-f016:**
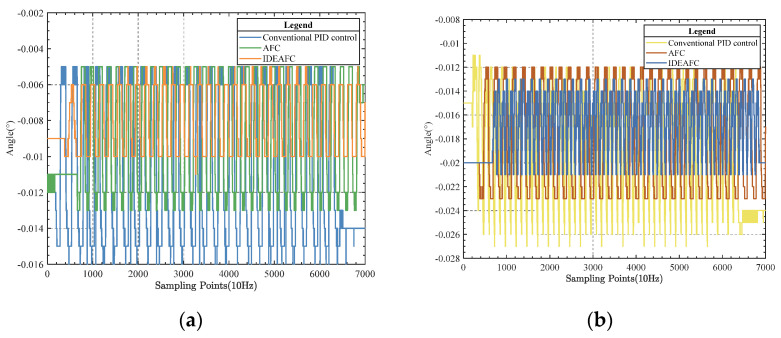
(**a**) Pitch angle data comparison; (**b**) roll angle data comparison.

**Figure 17 sensors-23-03172-f017:**
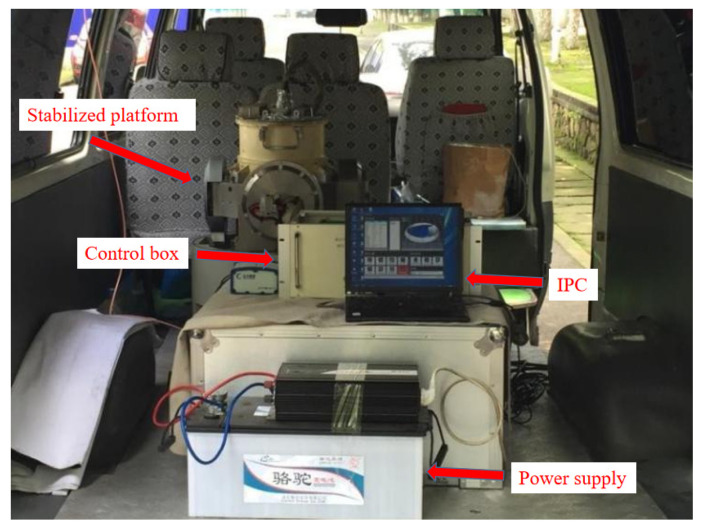
Vehicle experiment environment.

**Figure 18 sensors-23-03172-f018:**
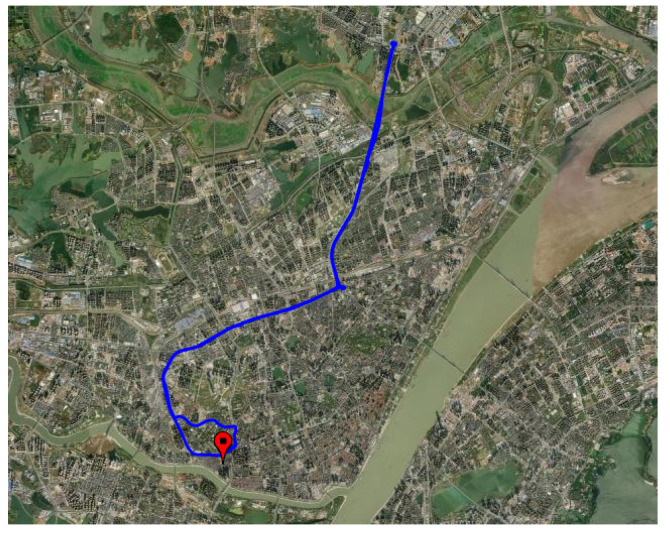
Vehicle experimentation route.

**Figure 19 sensors-23-03172-f019:**
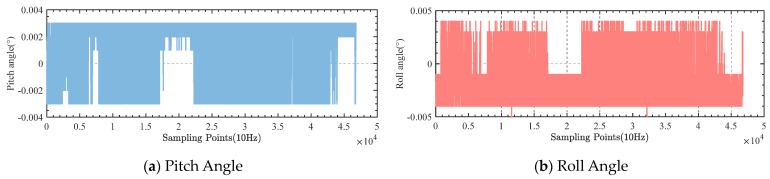
Vehicle experimentation results.

**Figure 20 sensors-23-03172-f020:**
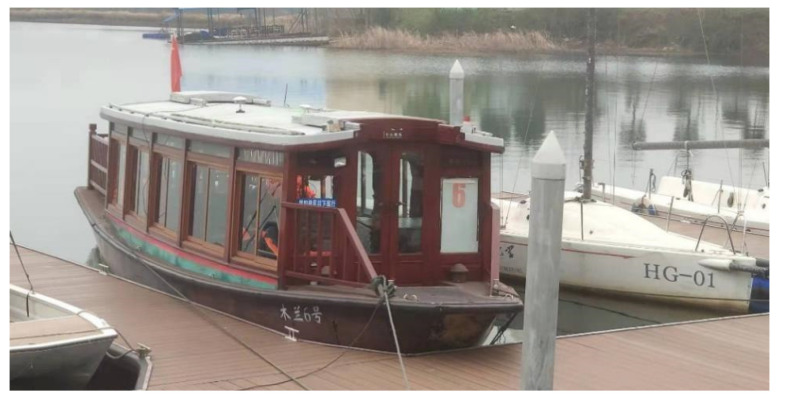
Experimental vessel.

**Figure 21 sensors-23-03172-f021:**
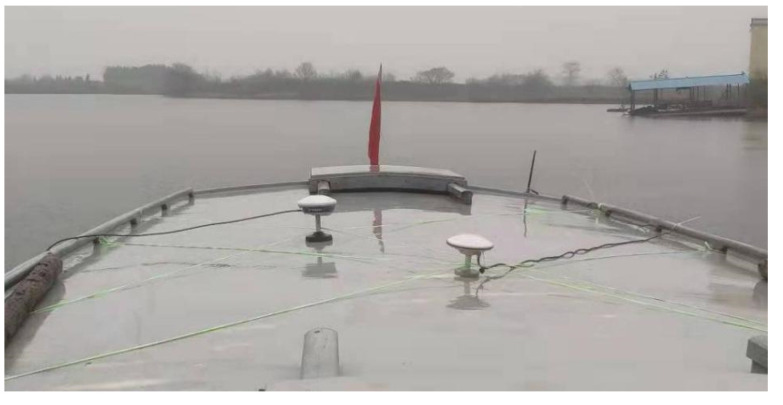
Installation position of GNSS antenna.

**Figure 22 sensors-23-03172-f022:**
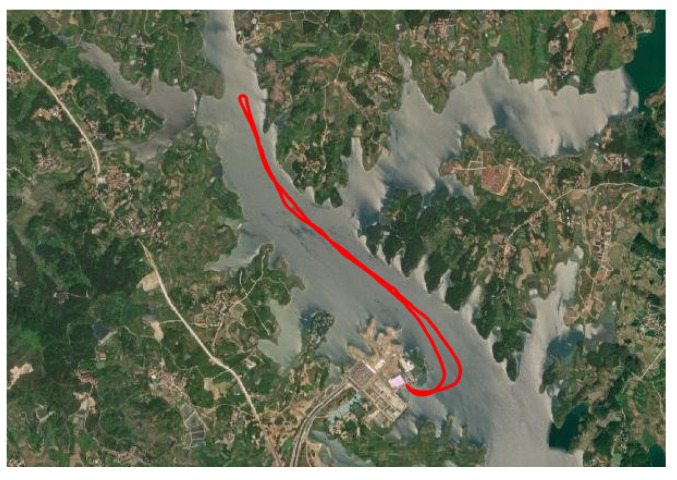
Navigation trajectory.

**Figure 23 sensors-23-03172-f023:**
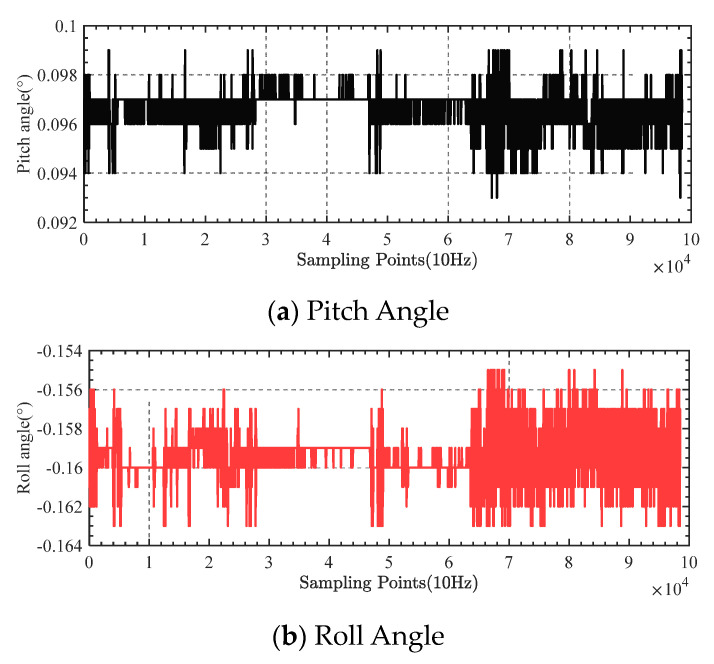
Ship-borne test results.

**Table 1 sensors-23-03172-t001:** Parameter setting.

Parameter Name	ADE-MMVP
Number of iterations	500
Population size	20
Individual Dimension	3
Code length	20
Scale factor λ	0.35
Variable factors	Max 1.8 Min 0.4
Crossover probability factor	Max 0.8 Min 0.2

**Table 2 sensors-23-03172-t002:** Comparison of control effects under static conditions.

	Pitch Angle Std	Roll Angle Std
Traditional PID control overall stability accuracy	2.98″	3.36″
AFC overall stability accuracy	2.56″	0.96″
IDEAFC overall stability accuracy	1.35″	0.72″
Improved relative to traditional PID control	54.7%	78.6%
Relative AFC improves	47.3%	25%

**Table 3 sensors-23-03172-t003:** Rolling period.

	Pitch	Roll
Gear 1	40 s	40 s
Gear 2	20 s	20 s
Gear 3	15 s	15 s

**Table 4 sensors-23-03172-t004:** Comparison of control effect under pitching conditions.

	Pitch Angle Std	Roll Angle Std
Traditional PID control overall stability accuracy	16.00″	17.27″
AFC overall stability accuracy	8.92″	12.23″
IDEAFC overall stability accuracy	7.28″	8.90″
Improved relative to traditional PID control	54.5%	48.5%
Relative AFC improves	18.4%	27.2%

**Table 5 sensors-23-03172-t005:** Comparison of control effects under dynamic rolling conditions.

	Pitch Angle Std	Roll Angle Std
Traditional PID control overall stability accuracy	15.61″	16.60″
AFC overall stability accuracy	11.92″	13.37″
IDEAFC overall stability accuracy	6.72″	9.82″
Improved relative to traditional PID control	60.0%	40.8%
Relative AFC improves	43.6%	26.6%

## Data Availability

The data that support the findings of this study are available from the corresponding author upon reasonable request.

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
