# Peer review of "An Improved Differential Evolution Adaptive Fuzzy PID Control Method for Gravity Measurement Stable Platform"

_sensors, 2023, doi:10.3390/s23063172_

Round 1

Reviewer 1 Report

The author has proposed a differential evolution adaptive fuzzy PID control algorithm for gravimeters that addresses several important problems such as stabilization accuracy of the gravimeter, mechanical friction, inter-device coupling interference, and nonlinear disturbances. The study is very interesting, and its effectiveness is verified by using both simulation and experiential results. However, authors need to address two important problems before publication.

(1) Paper is weak from a language point of view. It is better to proofread paper by a native speaker.

(2) There is a lot of published literature on the platform gravimeter system. The author must cite at least 20 to 25 papers.

Reviewer 2 Report

Dear Authors, 

I really appreciated your manuscript, clear and convincing. I just suggest two additions:

1/ a gravimeter is a pass-band instrument. You give only a requirement of global 4.8 minutes of arc for the platform misalignment tolerance. It is clear that this value depends in fact of the wavelenght of the gravity signal being investigated. Could you discuss this more in depth?

2/ Your article will gain more readership if you add a short description, in an annex,  about the family of ADE-MMVP algorithms. This is very cryptic for a lot of people, including me, and references are not enough.

I suggest to read the following paper, that discuss the inversion of airborne gravity data, in relation with their wavelength contents: 

Abbasi M. et al., Airborne LaCoste & Romberg gravimetry: a space domain approach, Journal of Geodesy, DOI 10.1007/s00190-006-0107-z, November 2006.

I also attach handwritten notes.

Reviewer 3 Report

The improved differential evolutionary adaptive fuzzy PID control proposed by the authors can achieve higher stability accuracy and could be applied to other control scenarios. Overall, I would like to suggest publishing the manuscript after minor revision, and a few issues should be carefully addressed before publication.

(1) Although it is not difficult to guess the meaning of mathematical symbols in equations (1) and (2), I would like to suggest the authors write down the definitions of all mathematical symbols shown in the text.

 (2) I would like to suggest the authors provide better presentations of the Simulation and Analysis section. Table 1 is redundant information, and can be deleted. Readers may be more interested in the design, flowchart, the set-up conditions and parameters of the numerical simulations, and how the simulations proceed. Roughly, readers can tell the superiority of IDEAFC from the comparisons in Figures 8 and 9, but a detailed quantitative analysis should be provided.

(3) Figure 11 suggests that the experiments are conducted in indoor environment, in which GNSS has poor performance. Would this affect the correction of the IMU’s drift and the outcome of the experiment?

(4) Why do the authors only provide the test of IDEAFC algorithm for the outdoor experiments including in-vehicle and shipboard experiments? Figure 17 and 18 are helpful, but don’t mean much.

What’s the driving speed, and how do the pitch and roll angles of the vehicle (or boat) change, during the entire experiment period? I would like to suggest the author add further interpretations on the variations of pitch and roll angle in Figures 19 and 20.
